# Similarities in Metabolic Flexibility and Hunger Hormone Ghrelin Exist between FTO Gene Variants in Response to an Acute Dietary Challenge

**DOI:** 10.3390/nu11102518

**Published:** 2019-10-18

**Authors:** Jessica Danaher, Christos G. Stathis, Matthew B. Cooke

**Affiliations:** 1School of Science, College of Science, Engineering and Health, Royal Melbourne Institute of Technology (RMIT) University, Melbourne VIC 3083, Australia; jessica.danaher@rmit.edu.au; 2Institute for Health and Sport, Victoria University, Melbourne VIC 3011, Australia; christos.stathis@vu.edu.au; 3Department of Health and Medical Sciences, Faculty of Health, Arts and Design, Swinburne University of Technology, Melbourne VIC 3122, Australia; 4Australian Institute for Musculoskeletal Science (AIMSS), Western Health, Melbourne VIC 3021, Australia

**Keywords:** FTO, metabolic flexibility, oral glucose load

## Abstract

The rs9939609 polymorphism of the fat mass and obesity-associated (FTO) gene has been associated with obesity, and studies have also shown that environmental/lifestyle interaction such as dietary intake might mediate this effect. The current study investigates the postprandial hormonal regulators of hunger and indirect markers of substrate utilisation and metabolic flexibility following a dietary challenge to determine if suppression of circulating ghrelin levels and/or reduced metabolic flexibility exist between FTO genotypes. One hundred and forty seven healthy, sedentary males and females (29.0 ± 0.7 yrs; 70.2 ± 1.1 kg; 169.1 ± 0.8 cm; 24.5 ± 0.3 kg/m^2^) complete a single experimental session. Anthropometric measures, circulating levels of active ghrelin, insulin and glucose, and substrate oxidation via indirect calorimetry, are measured pre-prandial and/or post-prandial. The FTO rs9939609 variant is genotyped using a real-time polymerase chain reaction. Metabolic flexibility (∆RER) is similar between FTO genotypes of the rs9939609 (T > A) polymorphism (*p* > 0.05). No differences in pre-prandial and/or postprandial substrate oxidation, plasma glucose, serum insulin or ghrelin are observed between genotypes (*p* > 0.05). These observations are independent of body mass index and gender. Altered postprandial responses in hunger hormones and metabolic flexibility may not be a mechanism by which FTO is associated with higher BMI and obesity in healthy, normal-weighted individuals.

## 1. Introduction

Genetic variations can predispose some individuals to be more susceptible to weight gain and obesity in the modern obesogenic environment [1,2]. Of particular interest has been the fat mass and obesity-associated (FTO) gene, with those homozygous for A-allele demonstrating an association with body mass index (BMI), waist circumference, type 2 diabetes (T2D), and other obesity-related traits [3,4,5]. While the mechanisms behind these associations are yet to be established, one reason could be due to higher food intake and diminished food satiety [6,7,8], with several lines of evidence supporting FTO’s role in regulating energy intake and eating behaviour [9,10].

Ghrelin, a gut “hunger hormone”, is a key mediator of appetite and subsequent body weight regulation. Recent evidence suggests that perturbations to circulating ghrelin levels and neural responses to food cues could explain FTO’s association with increased susceptibility to gaining weight. Karra et al. [11] showed that the FTO rs9939609 risk A-allele is associated with increased FTO expression, reduced N^6^-methyladenosine (m^6^A) ghrelin mRNA methylation and increased ghrelin expression. Increased expression and a subsequent rise in circulating acyl-ghrelin (AG) levels has been linked to greater food intake and a preference for energy-dense foods [12]. Recently, higher postprandial AG concentrations, which coincided with higher postprandial appetite and ad libitum energy intake, have also been evident in FTO rs9939609 risk A-allele compared to those homozygous to TT [13]. While the evidence suggests that FTO rs9939609 genotype impacts circulating AG levels [11], impaired central nervous system satiety processing could also be a factor contributing to reduced satiety and increased food intake [14].

FTO is also widely expressed in peripheral tissues involved in energy metabolism, including the skeletal muscle. Metabolic flexibility, the transition in fuel selection from fasting to fed state, is important in governing substrate utilisation in response to conditional changes in metabolic demand [15]. Metabolic inflexibility has been linked to the development of obesity and T2D and could provide another mechanism by which FTO is associated with increased susceptibility to becoming overweight and obese. An age-dependent decline in skeletal muscle FTO mRNA expression has been linked to peripheral defects in substrate metabolism in response to insulin infusion—namely, lower oxidation of glucose and higher oxidation of fat [16]. In type 2 diabetics, elevated FTO protein expression has been associated with reduced glucose oxidation and blunted suppression of fat oxidation in insulin-stimulated conditions [17]. It has been suggested that FTO’s influence on whole body energy homeostasis and the modulation of substrate oxidation could be via regulation from adenosine monophosphate activated protein kinase (AMPK) [18]. Recent findings by Wu et al. [18] have shown that activation of AMPK down-regulates FTO expression, whereas inhibition of AMPK up-regulates FTO expression in skeletal muscles. Regulation of FTO via AMPK subsequently effects m^6^A methylation levels, leading to changes in fat oxidation and lipid deposition [18]. Thus, changes in AMPK and FTO levels/function following acute perturbations in energy status and fuel selection could elucidate mechanisms of previously identified links between FTO expression and metabolic defects, which are possibly the result of metabolic inflexibility.

Therefore, the aim of the present study is to examine circulating levels of active ghrelin, substrate oxidation and metabolic flexibility between risk (AA) and non-risk allele (TT) genotypes of the rs9939609 FTO variant in normal-weight male and females in response to standard oral glucose test. We hypothesise that following the ingestion of the glucose load, individuals homozygous to risk A-allele will exhibit an attenuated post-prandial suppression of circulating ghrelin levels and reduced metabolic flexibility irrespective of BMI and gender.

## 2. Materials and Methods

### 2.1. Participants

A total of 150 sedentary males and females aged between 20–50 years were recruited for this study. Participants were excluded from participating if they had diagnosed diabetes (fasting blood glucose greater than 7.0 mmol·L^−1^), were performing any regular fitness training (>30 min, 3 × per week) for 6 months prior, were taking contraindicated prescription medication that influenced their metabolism (including thyroid, hyperlipidaemia, hypoglycaemic or antihypertensive) or were pregnant. Participants believed to meet the eligibility criteria were asked to provide written consent based on documents previously approved by the Victoria University Human Research Ethics Committee (*HRETH 12/197*), and all procedures were performed in accordance with the ethical standards set out in the 1964 Declaration of Helsinki.

Participants were asked to refrain from consuming caffeine and alcohol, and from undertaking strenuous exercise, 24 h prior to attending the experimental trial. Participants recorded their dietary intake over 4 days prior to the trial, and were asked to consume a carbohydrate predominant meal the night before with guidelines provided by a qualified nutritionist. Experimental trials were conducted in the morning, 10–12 h after the last meal, to ensure participants were in a pre-prandial/basal state.

### 2.2. Experimental Trial Protocols

#### Genotyping

Standard buccal swabs from the inside of the cheek were collected and subsequently used for DNA extraction using QuickExtract solution (Illumina, VIC, Australia). Each buccal swab was agitated in 500 µL of QuickExtract solution before being heated at 65 °C for 1 min and 95 °C for a following 2 min. Prior to subsequent analysis, samples were diluted with nuclease-free H_2_O to achieve a final concentration of approximately 7 ng·µL^−1^ DNA, as confirmed spectrophotometrically.

Genotyping of the rs9939609 (T > A) polymorphism of the FTO gene was performed via TaqMan allelic discrimination assay (Life Technologies, VIC, Australia). A total of 20 ng of DNA (~2.5 µL per sample) was added to each well with 0.5 µL ×20 TaqMan SNP Genotyping Assay and 5 µL TaqMan Genotyping Master Mix. Nuclease-free H_2_O was added to create a final well volume of 11 µL. Each sample underwent 50 amplifications on a CFX96 Real-Time thermal cycler (Bio-Rad Laboratories, VIC, Australia). For quality control purposes, a positive and negative control was used. Context sequence for the SNP tested was [VIC/FAM] GGTTCCTTGCGACTGCTGTGAATTT[A/T]GTGATGCACTTGGATAGTCTCTGTT. Florescent signals of the two probes were monitored throughout the entire amplification using CFX Manager 2.1 software (Bio-Rad Laboratories, VIC, Australia). FTO genotype was determined by graphic examination of fluorescent intensity. The overall genotyping efficiency was 98.0%, with results unable to be obtained in three participants, resulting in a total *n* = 147.

### 2.3. Anthropometric Measurements and Dietary Analysis

Total body mass, fat mass and muscle/lean body mass (LBM) were recorded using bioelectrical impedance scales (Tanita, VIC, Australia). Height, hip and waist circumference, and blood pressure, were measured using a stadiometer, tape measure and sphygmomanometer (Omron HEM7322, Omron Healthcare, VIC, Australia), respectively. Dietary records (4 day) were evaluated using Foodworks™ (Australia—Diet and Recipe Analysis (AusFoods), version 7, 2012, Xyris Software) to determine each participant’s average daily total energy intake, as well as the percentage contributions of protein, carbohydrate, fat, saturated fat (from total fat), alcohol and fibre.

### 2.4. Oral Glucose Load (OGL) Challenge

In order to examine glucose handling, participants performed a standard OGL challenge, which involved ingesting 75 g glucose in the form of a 300 mL solution (GlucoScan, SteriHealth, VIC, Australia). Participants were asked to consume the drink within a 2 min period for consistency (i.e., to standardise gastric emptying and intestinal absorption rates) between the subgroups.

### 2.5. Blood Sampling, Treatment and Analysis

Blood was sampled from a vein in the antecubital space, and the cannula was kept patent using isotonic saline (0.9% NaCl, Pfizer, NWS, Australia). A 20 mL blood sample was collected prior to OGL ingestion and at three time points post ingestion (30 min, 60 min and 120 min). Blood samples were collected using lithium heparin and serum-separating tubes (BD Vacutainer, BD Bioscience, NSW, Australia). Lithium heparin-treated blood was transferred into Eppendorf tubes and centrifuged at 12,000 rpm for 2 min, whilst serum-separating tubes were left to sit at room temperature for 30 min prior to centrifugation at 1200× *g* for 10 min. Plasma was decanted and immediately analysed for glucose concentration (YSI 2300 STAT; Yellow Springs Instruments, OH, USA). Serum was decanted and used to analyse serum ghrelin and insulin concentrations via a Ghrelin (Active) ELISA Kit (EMD Millipore, VIC, Australia) and Insulin ELISA Kit (Invitrogen, VIC, Australia), respectively. Insulin sensitivity index (ISI) was calculated using the composite index [19], and homeostasis model assessment of insulin resistance (HOMA-IR) was calculated as per Matthews et al. [20].

### 2.6. Respiratory Gas Exchange Sampling and Analysis

Respiratory data were collected prior to and for 60 min following OGL ingestion to determine metabolic flexibility via RER data. This involved participants wearing a respiratory mouthpiece and nose clip to have their inspired O_2_ and expired CO_2_ measured in 30 sec intervals via indirect calorimetry (Moxus, AEI Technologies, PA, USA). Metabolic flexibility was measured as the RER delta change (∆RER) 60 min post OGL ingestion compared to pre-prandial measurements. Substrate utilisation was calculated using commonly used stoichiometric equations [21], with the assumption that protein oxidation was minor and constant. Energy expenditure was calculated based on the following formula, with respiratory values in L·min^−1^ units:
Energy Expenditure (kJ·min^−1^) = 16.318 * VO_2_ − 4.602 * VCO_2_

### 2.7. Statistical Analysis

Results are expressed as mean ± SEM. Statistical analysis was performed using SPSS software (IBM SPSS Statistics for Windows, Version 20, NY, USA). When necessary, raw data were log-transformed to obtain normality. Two-way ANOVA’s with repeated measures were used to calculate individual significance in plasma, serum and indirect calorimetry data (energy expenditure, RER and substrate oxidation), with time as the within group factor and genotype as the between group factor. Where an interaction was detected, multiple comparisons with Tukey’s post hoc tests were completed to identify differences. One-way ANOVA’s were performed for participant characteristic and nutritional intake data, with unpaired t-tests completed when interactions between factors were identified. Sample power was examined by partial eta-squared (effect size) (ηp^2^), observed power (β) and power (1-β) multivariate analysis of plasma, serum and indirect calorimetry data. Linear regression and covariant analysis (ANCOVA) were used to determine the effect of BMI and gender on allelic representation of dependent variables. The level of probability was set at *p* < 0.05.

## 3. Results

Participant FTO genotype groups (rs9939609 variant alleles; AA, AT and TT) are presented in Table 1. Genotype frequency was consistent with Hardy-Weinberg equilibrium: 21.8% AA, 42.2% AT and 36.0% TT. No differences in age, total body mass, height, BMI, hip and waist circumference, fat mass, LBM or blood pressure were observed when participants were separated by FTO genotype (*p* > 0.05).

### 3.1. Nutritional Intake Analysis

Daily caloric intake was similar between FTO genotypes (*p* > 0.05). A genotype interaction was detected for percentage carbohydrate intake (*p* = 0.017) and percentage saturated fat (*p* = 0.002) and monounsaturated fat (*p* = 0.026) (from total fat) intake (Table 2). Subsequent analysis revealed that AA genotypes consumed a greater percentage of carbohydrates than AT genotypes (*p* = 0.037), a larger percentage of saturated fat than AT (*p* = 0.001) and TT genotypes (*p* = 0.004) and a lower percentage of monounsaturated fat than AT (*p* = 0.008) and TT genotypes (*p* = 0.027).

Daily intake of protein, total fat, polyunsaturated fat, alcohol, fibre as well as n-3 and n-6 fatty acids was similar among FTO genotypes (*p* > 0.05) (Table 2 and Table 3).

### 3.2. Substrate Utilisation and Metabolic Flexibility Analysis

A significant time effect for RER, energy expenditure (kJ·kgLBM^−1^·min^−1^) as well as glucose and fat oxidation (g·kgLBM^−1^·min^−1^) was observed post OGL (*p* < 0.001), regardless of FTO genotype. Subsequent pairwise comparisons revealed a significant increase in RER, energy expenditure and glucose oxidation from pre OGL to 60 min post OGL (*p* < 0.01). Conversely, fat oxidation was significantly reduced over the 60 min duration post OGL (*p* < 0.01).

No genotype main effect or genotype by time interaction was identified for RER, energy expenditure or oxidation of fat and glucose (*p* > 0.05) (Table 4). When adjusted for BMI and gender, energy expenditure and substrate oxidation remained similar between FTO genotypes (*p* > 0.05) (Appendix A).

Metabolic flexibility (∆RER) was similar among FTO genotypes (*p* = 0.299) (Figure 1). The absence of an effect of BMI (*p* = 0.32, R^2^ = 0.043) and gender (*p* = 0.357, R^2^ = 0.040) on metabolic flexibility response between FTO genotypes was confirmed by ANCOVA. When observing the influence of BMI regardless of FTO genotype, metabolic flexibility (∆RER) was found to be significantly slower in overweight individuals (BMI ≥ 25 kg/m^2^) than lean individuals (BMI ≤ 24.9 kg/m^2^) (*p* = 0.009).

Power analysis results for RER (ηp^2^ = 0.020; 1-β = 0.690), energy expenditure (ηp^2^ = 0.006; 1-β = 0.888), glucose oxidation (ηp^2^ = 0.007; 1-β = 0.876) and fat oxidation (ηp^2^ = 0.005; 1-β = 0.893) are available in Appendix A, Appendix A.

### 3.3. Plasma Analysis

A significant main effect for time was observed for plasma glucose concentration following glucose ingestion (*p* < 0.001), regardless of FTO genotype. Subsequent pairwise comparisons revealed a significant increase in plasma glucose from pre OGL to 30 min (*p* < 0.001), 60 min (*p* < 0.001) and 120 min (*p* < 0.05) post OGL.

No genotype main effect (*p* = 0.114) or genotype by time interaction (*p* = 0.427) was identified for plasma glucose (Figure 2). Plasma glucose concentrations remained similar between FTO genotypes when adjusting for BMI and gender (*p* > 0.05) (Appendix A).

### 3.4. Serum Analysis

Serum insulin and ghrelin concentrations were measured in a subset of 87 participants (*n* = 29 per FTO rs9939609 genotype). A significant main effect for time was observed for serum insulin and ghrelin concentrations post OGL (*p* < 0.001), regardless of FTO genotype. Subsequent pairwise comparisons revealed a significant increase in serum insulin from pre OGL to 60 min (*p* < 0.001) and 120 min (*p* < 0.001) post OGL, and a significant decrease in serum ghrelin from pre OGL to 60 min (*p* < 0.001) and 120 min (*p* < 0.001) post OGL.

No genotype main effect (SI, *p* = 0.439; SG, *p* = 0.626), or genotype by time interaction (SI, *p* = 0.330; SG, *p* = 0.979) was identified for serum insulin (Figure 3A) and serum ghrelin concentrations (Figure 3B). When adjusted for BMI and gender, serum insulin and ghrelin concentrations remained similar between FTO genotypes (*p* > 0.05) (Appendix A).

No difference in HOMA-IR (*p* = 0.382) or ISI (*p* = 0.828) was observed between genotypes. The absence of an effect of BMI and gender on HOMA-IR (BMI, *p* = 0.390, R^2^ = 0.110; Gender, *p* = 0.372, R^2^ = 0.027) and ISI (BMI, *p* = 0.824, R^2^ = 0.005; Gender, *p* = 0.822, R^2^ = 0.024) outcomes between FTO genotypes was confirmed by ANCOVA.

## 4. Discussion

Nutritional stimuli provide acute perturbations to the systemic hormonal milieu and skeletal muscle metabolic homeostasis. This may assist in unmasking differences between allelic variants of FTO in their ability to adapt, which would otherwise remain unnoticed under resting conditions. Our study demonstrates that, independent of body mass and gender, individuals carrying the FTO risk A-allele demonstrate similar changes in postprandial hormonal regulators of hunger and indirect markers of substrate utilisation and metabolic flexibility following an acute glucose load challenge. The study suggests that in healthy, normal-weighted males and females, altered postprandial responses in hunger hormones and metabolic flexibility may not be a mechanism by which FTO is associated with higher BMI and obesity.

Circulating levels of orexigenic hormones such as AG alter appetite and food intake and modulates brain activity within both homeostatic and reward-related brain regions [22]. Serum levels of ghrelin typically rise in the pre-prandial state and fall in the postprandial state—reflecting the hunger and satiety that precedes and follows eating. In the current study, serum ghrelin levels were elevated in the pre-prandial state following 8–12 h fasting and declined over a 2-h period following OGL ingestion. These changes were similar between FTO genotypes, with no significant differences between those homozygous for the risk allele (AA) and wild type (TT). Similarly, recent investigations by Goltz et al. [23] and Melhorn et al. [14] reported no gene-diet interaction, with comparable circulating ghrelin levels or ratings of hunger observed between FTO genotypes prior to and/or following a standardised meal in both lean, overweight and obese individuals. In contrast, Karra et al. [11] observed elevated basal ghrelin mRNA expression in peripheral blood cells and reduced suppression of circulating ghrelin in FTO risk A-allele carriers within a subset of 20 individuals. In a comparable cohort, Dorling and colleagues also found an attenuated AG suppression after a fixed meal, which coincided with higher ad libitum energy intake compared with TT [13]. However, similar to our study, they found no difference in fasting concentrations of AG between genotypes, but when presented as a ratio of AG/des-acyl-ghrelin (DAG), AA genotype demonstrated higher levels compared to TT [13]. Interestingly, the Dorling et al. [13] study also observed lower fasting butyrylcholinesterase (BChE) activity in carriers of AA compared to TT, which may offer a potential explanation for the reported higher AG:DAG ratio and energy intake between genotypes. BChE activity increases AG hydrolysis in plasma, leading to greater DAG and a lower AG:DAG ratio, which has been linked to lower energy consumption and lower adiposity in mice [12]. A limitation in the current study is that BChE activity was not measured, and thus while AG levels were similar between genotypes, the AG:DAG ratio may have been different due to the activity of BChE.

The FTO risk A-allele has previously been associated with a greater preference for, and consumption of, energy-dense foods [24,25]. In the current study, daily total energy consumption and total consumption of fat were similar between genotypes, which is supported by other studies showing no difference in caloric intake between FTO rs9939609 genotypes [26,27]. However, AA genotypes were found to consume a higher percentage of carbohydrates compared to AT genotypes, a larger percentage of saturated fat compared to AT and TT genotypes and a lower percentage of monounsaturated fat compared to AT and TT genotypes. While metabolic adaptation can occur in response to long-term macronutrient intake [28], we found no differences in baseline substrate utilisation or carbohydrate and/or fat oxidation, and thus any differences in long term habitual macronutrient intake are unlikely to have impacted the study findings. Given the dietary, nutritional and genomic implications of long-chain polyunsaturated fatty acids (LCPUFAs) such as *n*-3 LCPUFAs intake [29,30], the current study analysed dietary omega 3 and 6 levels between genotypes. No significant differences were observed, and thus any influence from such fatty acids in response to the OGL is likely to be similar between genotypes. It is important to acknowledge the complexities of food intake and food preference, with many determinants influencing eating behaviours at household and population levels. The validity and reliability of energy intake and food composition variables may be limited due to the short-term and self-reporting nature of data collection and could be compounded by the relatively small cohort examined.

Within the skeletal muscle, FTO influences energy sensing pathways and substrate utilisation [31]. Recent findings suggest AMPK, via FTO-dependent demethylation of N^6^-methyladenosine (m^6^A), regulates lipid oxidation and accumulation in skeletal muscle cells [18]. Furthermore, an age-related decline in FTO muscle expression is associated with a shift from glucose to fat oxidation [16]. It is clear that FTO is regulated by nutritional status, with caloric input and composition of macronutrient potentially changing FTO gene expression levels. However, whether individuals homozygous to the FTO risk A-allele display altered peripheral metabolic regulation in response to an acute shift in energy demand is unclear. The current study observed no significant difference in RER (an indirect marker of metabolic flexibility) over a 1-h monitoring period following glucose ingestion independent of gender and BMI. While no other study to our knowledge has investigated the relationship between metabolic flexibility and FTO rs9939609 polymorphism, previous investigations have reported no associations between energy expenditure and gene variants of FTO in either a pre-prandial [6,16,32] or post-prandial state [32,33]. Additionally, investigations to date have found no association between the FTO rs9939609 polymorphism and the ability to regulate substrate oxidation under resting conditions or in response to nutritional stimuli in an obese [34] and elderly cohort [16]. It is possible that the apparent absence of differences in metabolic responses between FTO allelic variants in the current study may be due to the one dimensional nature of the glucose-stimulated challenge. In other words, an acute manipulation of glucose oxidation and storage pathways may be insufficient to instigate changes in FTO function or display potential genotypic differences. Conversely, longer-term, repetitive dietary exposure may be necessary to influence changes in FTO levels and its demethylase function. Elevated basal levels of FTO in the skeletal muscle of individuals with the lifestyle disease T2D [17] may support this concept, with changes in expression levels potentially reflecting long-term exposure to adverse dietary choices (e.g., excessive intake of simple sugars) and/or metabolic disturbances (i.e., an impaired ability to utilise glucose and thus a depletion of nutrients). Measuring skeletal muscle FTO level and/or function was not within the scope of the current study, thus it is unknown whether differences in FTO protein level and/or activity existed in the current cohort.

A number of limitations do exist in the current study. Firstly, we studied males and females that were young, healthy and of normal-weighted range. Thus whether the responses observed would be evident in other populations such as older adults, and in cohorts overweight and obesity, could not be determined. However, this may also be a strength of the study, as it was relatively narrow in its participant characteristics and thus reduced the within-subject variation in response to the acute glucose ingestion. Secondly, it is clear that whole body impairment in glucose control was not measured in the current study and thus we cannot comment on defects in insulin signalling in adipose tissue. However, while adipose tissue is an active component of glucose regulation, it is likely that other peripheral tissues such as skeletal muscle, which accounts for up to 80% of glucose clearance under insulin-stimulated conditions [35], is a major contributor to glucose control. Thirdly, the inclusion of a condition where no intervention takes place did not occur, and thus any natural oscillation in the outcomes due to individual differences in rates of stomach distention and gastric emptying as well as differences in gut microbiota could not be quantified and, therefore, the “true” effect of the intervention could not be assessed [23,36]. Fourthly, the flexibility of skeletal muscle to switch between oxidizing fat and glucose is related to insulin sensitivity, glycogen stores, percent body fat, fitness levels and long-term habitual dietary intake [37]. The current study controlled for as many of these influencing factors as possible, with physical activity and the evening meal before the OGL kept as stable as possible prior to the testing day. Furthermore, the non-significant differences in body weight, body fat percentage and LBM between genotypes and secondary analyses performed factoring in body weight suggest little influence of these factors on the observed results. Finally, the probability of type II error was found to be high based on power analysis of the respiratory data, suggesting that a greater sample size may be required to detect potential dietary-induced changes to respiratory gases between FTO genotypes. However, when participants were separated based on BMI, independent of FTO genotype, a significant reduction in metabolic flexibility was evident in those with a BMI of ≤24.9 kg/m^2^ compared to those with a BMI of ≥25.0 kg/m^2^ (data not shown). This indicates that the timeframe of the dietary challenge was sufficient to detect differences in metabolic flexibility.

## 5. Conclusions

Following an acute dietary challenge within healthy, normal-weighted male and females, postprandial responses in serum glucose and insulin, hormonal regulators of hunger and indirect markers of substrate utilisation and metabolic flexibility were not significantly influenced by genotypes of FTO rs9939609 polymorphism. However, future research should adopt more sensitive measurements of metabolic change within blood and skeletal muscle (i.e., metabolomics) and/or utilise greater physiological stress (i.e., exercise) or multiple stressors (i.e., chronic exposure), to assist in potentially unmasking FTO’s role in energy metabolism. In conclusion, altered postprandial responses in hunger hormones and metabolic flexibility may not be a mechanism by which FTO is associated with higher BMI and obesity in healthy, normal-weighted individuals.

## Figures and Tables

**Figure 1 nutrients-11-02518-f001:**
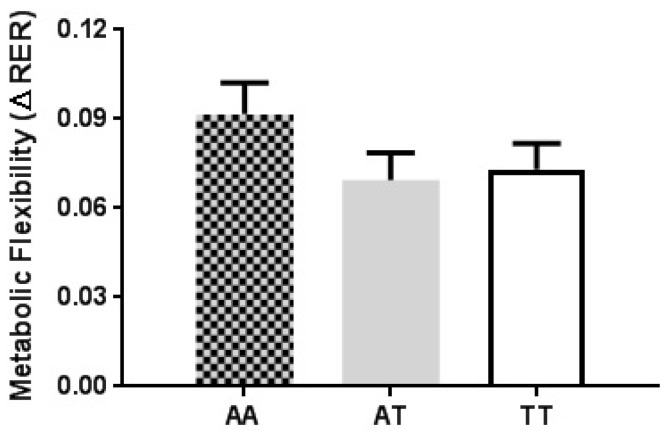
Metabolic flexibility measured as the ∆RER 60 min post OGL compared to pre OGL, with participants separated on their FTO rs9939609 genotype. Values expressed as mean ± SEM.

**Figure 2 nutrients-11-02518-f002:**
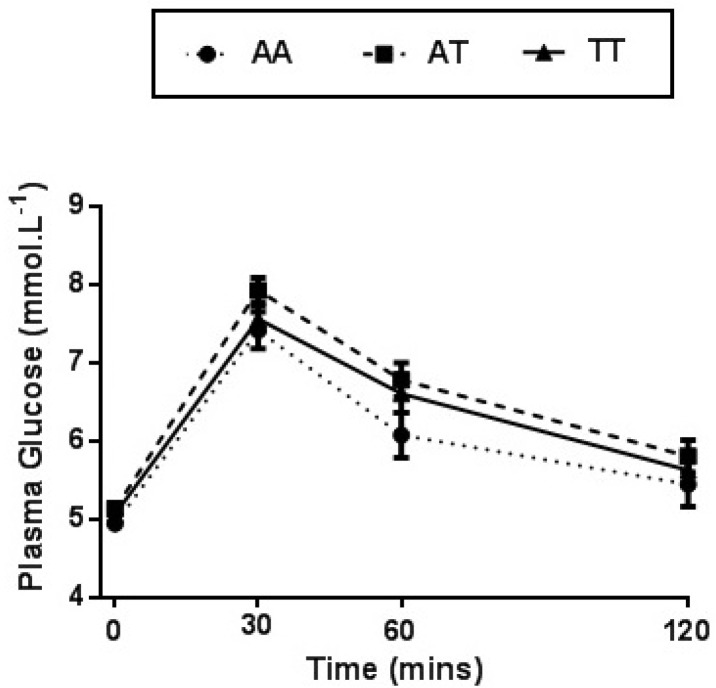
Plasma glucose concentration observed prior to and following an OGL challenge between genotypes for the FTO rs9939609 polymorphism. Values expressed as mean ± SEM. The main effect for time was observed in response to an OGL, *p* < 0.001.

**Figure 3 nutrients-11-02518-f003:**
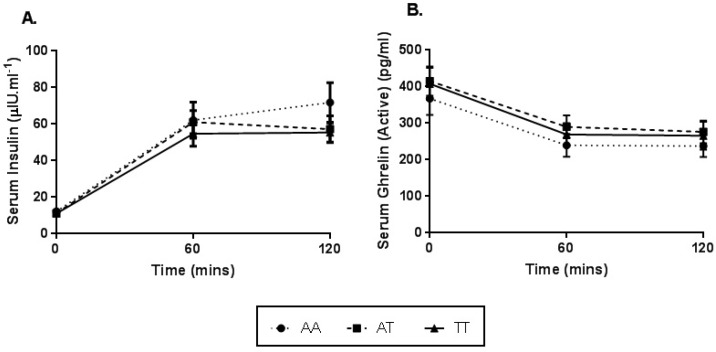
Serum insulin (**A**) and ghrelin (**B**) concentrations observed prior to and following an OGL challenge between genotypes for the FTO rs9939609 polymorphism. Values expressed as mean ± SEM. A main effect for time was observed in A and B response to an OGL, *p* < 0.001.

**Table 1 nutrients-11-02518-t001:** Clinical characteristics of the studied cohort based on their fat mass and obesity-associated (FTO) rs9939609 genotype. Values are expressed as mean ± SEM.

Genotype	AA	AT	TT	*P Value*
*n*	32	62	53	
Age (year)	29.8 ± 1.7	28.8 ± 1.0	28.9 ± 1.2	0.839
Total Body Mass (kg)	70.7 ± 2.3	70.3 ± 1.5	69.9 ± 2.1	0.971
Height (cm)	169.0 ± 1.8	169.7 ± 1.2	168.3 ± 1.3	0.713
BMI (kg/m^2^)	24.6 ± 0.6	24.5 ± 0.5	24.5 ± 0.6	0.981
Hip Circumference (cm)	102.1 ± 1.6	101.1 ± 0.9	100.9 ± 1.2	0.804
Waist Circumference (cm)	77.6 ± 1.7	77.0 ± 1.1	77.1 ± 1.4	0.958
Fat Mass (%)	27.5 ± 1.6	26.3 ± 1.2	27.4 ± 1.2	0.769
Lean Body Mass (kg)	48.1 ± 1.8	49.0 ± 1.3	47.9 ± 1.5	0.830
Systolic BP (mmHg)	121.9 ± 2.3	122.8 ± 1.1	122.8 ± 2.1	0.945
Diastolic BP (mmHg)	76.0 ± 1.6	72.7 ± 1.2	74.7 ± 1.2	0.190

**Table 2 nutrients-11-02518-t002:** Nutritional intake analysis of the studied cohort based on their FTO rs9939609 genotype. Values are expressed as mean ± SEM. Significant interactions (bolded, *p* < 0.05) were observed between FTO genotypes.

Genotype	AA	AT	TT	*P Value*
Total Energy (kcal)	1779.4 ± 79.7	1850.2 ± 63.0	1808.4 ± 61.2	0.768
Protein (%)	19.9 ± 0.9	22.1 ± 0.7	20.7 ± 0.6	0.089
Total Fat (%)	34.0 ± 2.4	31.8 ± 0.8	32.8 ± 0.9	0.460
Saturated Fat (%)	43.9 ± 1.4	37.0 ± 1.0	37.9 ± 1.4	0.002
Monounsaturated Fat (%)	41.0 ± 0.8	44.8 ± 0.7	44.2 ± 1.0	0.026
Polyunsaturated Fat (%)	15.1 ± 0.9	18.2 ± 0.8	17.83 ± 1.0	0.065
Carbohydrate (%)	44.0 ± 1.3	40.9 ± 0.9	41.6 ± 1.0	0.017
Alcohol (%)	0.8 ± 0.3	1.8 ± 0.4	1.7 ± 0.4	0.237
Fibre (%)	2.4 ± 0.1	2.3 ± 0.1	2.4 ± 0.1	0.811

**Table 3 nutrients-11-02518-t003:** Average daily fatty acid intake of the studied cohort based on their FTO rs9939609 genotype. Values are expressed as mean ± SEM.

Genotype	AA	AT	TT	*P Value*
Alpha-Linolenic Acid (ALA, 18:3 n-3) (g)	1.29 ± 0.19	1.41 ± 0.11	1.53 ± 0.17	0.604
Eicosapentaenoic Acid (EPA, 20:5 n-3) (g)	0.05 ± 0.01	0.12 ± 0.03	0.12 ± 0.03	0.262
Docosapentaenoic Acid (DPA, 22:5 n-3) (g)	0.07 ± 0.02	0.09 ± 0.01	0.09 ± 0.01	0.733
Docosahexaenoic Acid (DHA, 22:6 n-3) (g)	0.20 ± 0.05	0.16 ± 0.03	0.16 ± 0.03	0.805
Linoleic Acid (18:2 n-6) (g)	8.68 ± 0.66	11.25 ± 0.80	10.18 ± 0.83	0.149

**Table 4 nutrients-11-02518-t004:** Respiratory exchange ratio (RER), and calculated energy expenditure (EE), glucose and fat oxidation pre oral glucose load (OGL) (0 min) and 60 min post OGL, with participants separated according to their FTO rs9939609 genotype. Values are expressed as mean ± SEM.

Genotype	AA		AT		TT		*P Value*
	Pre OGL	Post OGL	Pre OGL	Post OGL	Pre OGL	Post OGL	
RER	0.82 ± 0.01	0.93 ± 0.01	0.80 ± 0.02	0.92 ± 0.01	0.81 ± 0.01	0.92 ± 0.01	0.251
Energy Expenditure (kJ·kgLBM^−1^·min^−1^)	0.102 ± 0.002	0.114 ± 0.003	0.104 ± 0.002	0.117 ± 0.002	0.104 ± 0.002	0.115 ± 0.002	0.679
Glucose Oxidation (g·kgLBM^−1^·min^−1^)	2.47 ± 0.28 × 10^−3^	4.08 ± 0.22 × 10^−3^	2.16 ± 0.18 × 10^−3^	4.16 ± 0.15 × 10^−3^	2.31 ± 0.21 × 10^−3^	4.13 ± 0.13 × 10^−3^	0.630
Fat Oxidation (g·kgLBM^−1^·min^−1^)	1.54 ± 0.10 × 10^−3^	1.15 ± 0.06 × 10^−3^	1.71 ± 0.08 × 10^−3^	1.19 ± 0.06 × 10^−3^	1.64 ± 0.10 × 10^−3^	1.15 ± 0.05 × 10^−3^	0.696

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
