# Peer review of "Similarities in Metabolic Flexibility and Hunger Hormone Ghrelin Exist between FTO Gene Variants in Response to an Acute Dietary Challenge"

_nutrients, 2019, doi:10.3390/nu11102518_

Round 1

Reviewer 1 Report

The manuscript submitted by Danaher et al. Studies the effects of a particular polymosphism on metabolic flexibility and the hunger hormone (ghrelin). In this regard, the authors address an important area related to nutrigenomics, especially the effects of a given stimulus (glucose) and the response to hunger.
The title, introduction, methodology, results and discussion are appropriate. However I have the following comments.

Major Comments:
1. The introduction uses too many references. 27 references for a study of these characteristics, is a lot. I suggest reducing 5 references.
2. The discussion adequately considers the different results. However, the genetic variants allow to explain all the observed results ?. What effect does weight and body fat have? What effects does the intake of specific foods have? This point should be included a small paragraph of the discussion.
3. Considering the observed results, what could be happening with the intestinal microbiota?
4. It would be very good for the study to include the intake data of n-3 and n-6 ​​PUFA in the subjects. Considering, all the dietary, nutritional and genomic implications in which these fatty acids participate. Particularly obesity.

Suggested References:

Valenzuela & Videla. The importance of the long-chain polyunsaturated fatty acid n-6 ​​/ n-3 ratio in development of non-alcoholic fatty liver associated with obesity. Food Funct 2011; 2: 644-8.

Echeverría et al., Long-chain polyunsaturated fatty acids regulation of PPARs, signaling: Relationship to tissue development and aging. Prostaglandins Leukot Essent Fatty Acids. 2016; 114: 28-34.

Minor comments:
1. Improve the wording of the study objective
2. Some phrases or sentences are very long, correct this problem
3. Some references should be deleted by the authors.

Author Response

We would like to thank reviewer 1 for their time reviewing the manuscript. Please see the attachment for authors' response.

Reviewer 2 Report

In the manuscript "Similarities in metabolic flexibility and hunger hormone ghrelin exist between FTO gene variants in response to the acute glucose-stimulated dietary challenge", the authors present their study examining the altered postprandial responses in hunger hormones and metabolic flexibility between FTO gene variants. The authors conclude that postprandial responses in serum glucose and insulin, hormonal regulators of hunger and indirect markers of substrate utilization and metabolic flexibility were not significantly influenced by genotypes of FTO rs9939609 polymorphism. The study is quite interesting. The methods are described clearly and the figures and tables are easy to understand the results.

Author Response

We would like to thank reviewer 2 for their time reviewing the manuscript.

Round 2

Reviewer 1 Report

The manuscript can be accepted in the current version. The authors made all the suggested modifications.